# The Interface Between Veterinary and Behavioral Management of Chimpanzees (*Pan troglodytes*) in a United States Sanctuary Demonstrated by Two Clinical Cases

**DOI:** 10.3390/vetsci11110523

**Published:** 2024-10-28

**Authors:** Raven Jackson, Rebekah Lewis, Amy Fultz

**Affiliations:** Chimp Haven, Keithville, LA 71047, USA; rlewis@chimphaven.org (R.L.); afultz@chimphaven.org (A.F.)

**Keywords:** clinical care, case reports, behavioral management, collaboration, welfare, metrics, health

## Abstract

Chimp Haven, the world’s largest chimpanzee sanctuary, houses over 300 chimpanzees with a variety of health needs including obesity, cardiac disease, and wounding. The veterinary and behavioral staff work together to develop individualized plans for the physical and behavioral well-being of the chimpanzees in their care. We specifically focus on the interface between the veterinarian and the behavioral program director via two example cases to illustrate our methodology. These insights depict a comprehensive understanding of the operational landscape and the strategic considerations for delivering exceptional and individualized care to chimpanzees housed in group settings.

## 1. Introduction

The culmination of a vision between chimpanzee veterinarians and primate behaviorists in the research setting, Chimp Haven became the first federal chimpanzee sanctuary for chimpanzees being retired from biomedical research in the United States in 2000 [1]. Chimp Haven has transported and cared for over 500 chimpanzee residents since that time. Currently housing 300 plus chimpanzees, our facility serves as the largest chimpanzee sanctuary in the world. The sanctuary is home to chimpanzees with a wide variety of individual health needs and age ranges (7 years to over 60); 70% of the colony has chronic health conditions with 56% of the population having comorbidities that require long-term care throughout their retirement years. Chimpanzees live in large, mixed-sex social groups within open-top corrals, multi-dimensional enclosures, and naturally forested habitats. Residents are currently in 30 social groups with an average group size of 10 individuals. Chimp Haven features some of the largest groups of captive chimpanzees, but this form of social housing presents challenges for both the veterinary and behavioral staff with monitoring and balancing individual vs. group needs. Social housing is an essential part of chimpanzee welfare, and although it comes with risks, well thought out behavioral and veterinary programs can assist in lessening these risks while promoting the overall well-being of the chimpanzees.

Our staff members and leaders have formulated a robust animal care program, integrating the efforts of the veterinary, behavioral, and husbandry teams. This collaborative teamwork facilitates social housing and voluntary participation in medical and behavioral management among the chimpanzees in our care. The veterinary team consists of an attending veterinarian, a clinical veterinarian, an animal training coordinator, and five technicians responsible for the medical care of the chimpanzees. Similarly, the behavioral team is composed of a behavior program director, a team of three positive reinforcement training program staff, and five behavioral wellness and enrichment technicians, primarily focused on executing the behavioral management plan. The husbandry department is a 31-member team that, in addition to providing foundational care such as cleaning, feeding, and shifting of the chimpanzees, plays an integral role in supporting the behavioral and veterinary programs. All these departments align their efforts to empower the chimpanzees with the ability to make choices and actively participate in their own care. This team approach allows for expertise in multiple areas as well as extensive communication between the teams [2].

Chimp Haven staff collects, collates, and shares summarized data with staff at weekly to monthly intervals, at the individual, group, and population level focusing on diverse aspects of chimpanzee welfare. This includes data on the chimpanzees’ well-being, such as wounding information gathered by our veterinary team, data related to introductions and social networks [3], as well as comprehensive behavior time budgets collected by the behavior and husbandry teams. Wounding is not an uncommon occurrence in group-housed chimpanzees; therefore, monitoring its severity and quantifying wounding frequency is important for group housing decisions and chimpanzee well-being. In addition, our organization integrates assessments of, and interventions for abnormal behavior into our data-sharing process. Abnormal behaviors such as self-injurious behavior, also referred to here as self-directed wound picking, are a priority for interventions from both the veterinary, husbandry, and behavioral staff. Through this systematic approach, we ensure that the entire animal care team is well-informed and equipped to address the evolving needs of the population.

The continuous and detailed decision-making and problem-solving process between veterinarians and behaviorists is critical for enhancing clinical management and improving chimpanzee welfare [4,5]. We employ this extensive communication and collaborative plan to advance environmental and facility design, including the utilization of chimpanzee medical interaction rooms for awake diagnostics, to develop individual care plans, and for the application of welfare assessment tools. Moreover, it is implemented in the operant conditioning program, or positive reinforcement training program, which plays a pivotal role in facilitating voluntary medical procedures without social isolation [6].

This approach not only underscores the commitment to the well-being of the chimpanzees but also reflects a comprehensive and integrated animal care strategy that considers both the medical and behavioral aspects [7]. Ultimately, this approach promotes the overall welfare of the chimpanzees, establishing it as a staple of care. To illustrate this approach, the authors have highlighted two specific cases where strategies were implemented that involved communication and coordination between the behavior program director and the veterinarian.

## 2. Materials and Methods

Chimp Haven’s animal care teams collect information on the chimpanzees in our care to inform data driven welfare decisions. We utilize multiple methods of data collection including but not limited to focal continuous observations, instantaneous and all occurrence observations [8], staff surveys, and anecdotal input from animal care staff. The following metric programs were utilized in the two case studies explored in this manuscript.

Metrics Utilized

Wound reporting: veterinarians provide a caseload report of the wounds and their severity that chimpanzees within the colony have received during a given week. Wound severity is scored on a 5-point scale as follows: a severity 1 wound is a superficial wound with a partial skin break; a severity 2 wound is a shallow cut with a full skin break; a severity 3 wound is a moderate wound less than one inch deep, but may also include crushing wounds or bruises; a severity 4 wound is a deep wound more than one inch deep; and finally, a severity 5 wound is a severe gaping wound or possibly a missing or amputated body part. This information is then transcribed into a single master Excel Spreadsheet by the behavior team. The Microsoft Excel file is then summarized via Microsoft Power BI to show the frequency and severity of wounding in individuals, groups, and the entire colony via graphs and tables shared with animal care team members.

Abnormal behavior reporting: this informal reporting system involves animal care staff reporting all observed occurrences of abnormal behavior via email to the behavior program director and veterinarian. Chimp Haven’s overall ethogram for chimpanzee behavior includes 36 possible abnormal behaviors (Appendix A). Self-directed wound picking, self-injurious behavior, and stereotypical locomotion are some of the abnormal behaviors that may be reported.

Social Monitoring System (SMS): this is a colony-wide observation program where staff gather baseline activity budget data of chimpanzee residents (both as individuals and within their social groups) to inform data-driven decisions regarding their welfare and management. All social groups are observed on an annual basis for at least six weeks. Other reasons to initiate additional SMS observations include introduction; concerns for wounding, aggression, or welfare; loss of a group member; new arrivals to the sanctuary; or release into novel enclosures. Observations are ten minutes long with one-minute focal scans. These observations are completed on a randomized schedule seven days a week between 7 a.m.–4 p.m.

Nearest Night Neighbor (NNN): this is an observation protocol that is enacted at various times including when chimpanzees first arrive at the sanctuary, after introductions, or if there is concern about social dynamics within a group. These formal observations take place during routine nighttime checks and involve scanning each individual chimpanzee in a group and recording their proximity to others.

Targeted Individual Assessment (TIA): in cases of extreme abnormal behavior, severe or repeated wounding, notable group aggression, and apparent behavioral changes, animal care managers, including the veterinarian and behavior program director, may recommend a TIA for an individual chimpanzee. These daily, continuous fifteen-minute observations are conducted by the behavioral team to gather as much information about an individual as possible in a short amount of time. TIA observations may occur at randomized times or may be tailored towards a specific event; for example, if there is concern about access to resources, observations would be centered around times of resource provision.

Positive Reinforcement Training (PRT): staff works with the chimpanzees in this training program aimed to assist with daily management procedures, including shifting chimpanzees from area to area, cooperative training for equitable feeding and enrichment opportunities, necessary veterinary procedures including training for body part presentation for wound and mobility assessments, medication compliance, and advanced medical behaviors such as laser (photo biomodulation) therapy to promote wound healing [9,10]. Depending on the goal of training, chimpanzees may be engaged multiple times a week in sessions with staff. Medical training for acute cases, including laser therapy and wound management, may occur daily, dependent on the veterinary prescription.

Housing and Care

Chimp Haven offers a diverse range of enclosures that cater to the individual needs of the chimpanzees. The enclosures include outdoor spaces with different flooring and substrate options such as grass, pine straw hay, and wood wool, open-air corrals, and multi-acre forested habitats with elevated climbing structures and wire shelves.

Connectivity between areas is facilitated by multiple overhead mesh chutes, enabling rotation throughout the sanctuary. This design allows the chimpanzees to have access to adjacent indoor bedrooms and outdoor areas, except when maintenance or inclement weather necessitates restriction. The sanctuary ensures the chimpanzees receive nesting materials daily and are provided a diet of fresh produce and commercially available primate diet (Mazuri 5NAA, Brentwood, MO, USA). Enrichment activities are offered every day, and the chimpanzees receive added forage several times a week.

Chimp Haven’s Preventative Health Program includes routine health monitoring at a frequency determined by the veterinarian(s) with no more than a three-year interval. These routine exams incorporate laboratory testing for the purpose of evaluating current health status. Chimpanzees are brought in for non-routine exams at the discretion of the veterinarian(s) when warranted by physical or behavioral symptoms that indicate the need for further veterinary care.

Chimpanzees at Chimp Haven are maintained in accordance with the United States Department of Agriculture Animal Welfare Act [11], Guide for the Care and Use of Laboratory Animals [12], and the Standards of Care for Chimpanzees Held in the Federally Supported Sanctuary System [13], among other regulations. The facility is fully accredited by the Association for Assessment and Accreditation of Laboratory Animal Care, International, and the Global Federation of Animal Sanctuaries.

The Chimpanzees

Subject 1

At the time of data collection, Subject 1 was a 34-year-old male chimpanzee that had a clinical history of arthritis, self-injurious behavior, and staph infections in his right leg. Subject 1 was transferred to Chimp Haven in 2006 after retiring from a biomedical research facility in Louisiana. He was maintained in a large 15-member multi-sex social group (nine females; six males). Data collection occurred from August 2019 to March 2022 depending on the protocol.

At the time of arrival at Chimp Haven in 2006, Subject 1 was an 18-year-old male. His right leg had scar tissue from prior staph infections. When the infection occasionally presented with worsening symptoms, he was managed with pharmaceutical interventions including antibiotics as needed, antihistamines, non-steroidal anti-inflammatory medications (NSAIDs), selective serotonin reuptake inhibitors (SSRIs), and medications for nerve pain. Subject 1 was the dominant male in his group for 13 years. In late summer of 2019, Subject 1’s group of 15 was integrated with three males and three females. For a variety of reasons, the veterinarian, behavior program director, and animal care staff determined that the new group was partially incompatible, and two males and one female from the introduced group were removed and reintegrated into another group in December 2019. During the following year, Subject 1 began to increase his self-directed picking of his own and others’ wounds; then, in October 2020, he received a serious wound from a groupmate. As the year progressed into spring 2021, Subject 1’s abnormal wound picking increased substantially [14].

Subject 2

Subject 2 was a 33-year-old male chimpanzee that had previously been experimentally used in monoclonal antibody testing. He had a clinical history of moderate cardiac disease that was maintained with antiarrhythmic medication. He had a behavioral history of depilation and consumption, self-aggression, and wound picking. He was transferred to Chimp Haven in 2019 after retiring from a biomedical research facility located in Texas.

Subject 2 arrived at Chimp Haven at 28 years old as a part of a trio (two males; one female); he and his groupmates were subsequently introduced to two other females in 2019, five more group mates in 2021 (three males; two females), and an additional female in 2022 creating a group of 11. As his group size increased, his social standing rose within the social hierarchy until he became the alpha male of his group. Although prone to loud displays, he was an affiliative alpha, taking time to build the necessary social bonds within his group and mediating disputes between others. In May 2023, a male and female pair were introduced to the group, and he was affiliative and accepting of the new pair. However, during the summer of 2023, a female within the existing group became aggressive towards a geriatric male, and she and another female were removed from the group. Although this improved the situation for the geriatric male, it caused social changes within the hierarchy. During the fall of 2023, Subject 2 did not have as many female supporters within his group. As the year progressed the new male began to challenge him for dominance.

## 3. Results

Subject 1

Subject 1 was monitored in various ways between 2019 and 2022 via abnormal behavior reports, SMS, NNN observations after the introduction, and wound reports. To begin with, between October 2019 and March of 2020, Subject 1’s group was monitored for social dynamic changes both during the day and at night following the introduction. One metric Chimp Haven uses to follow chimpanzees after introductions is whether or not they are comfortable coming inside at night. Chimp Haven provides the chimpanzees with the choice to sleep inside or outside except in cases of inclement weather. Subject 1 spent 90.97% of his nights inside between August 2019 and March 2020, and also spent time near multiple group members, both from his old group and in the new group, during this timeframe (Figure 1).

Subject 1’s time budget at this time indicated that he was spending 10.91% of his behavioral time budget engaged in affiliative behaviors (Figure 2), which was much higher than the colony overall, and very little time on aggressive or submissive behaviors (0.45% and 0.91%, respectively). Abnormal behavior accounted for 3.18% of his activity during this timeframe according to this metric. The abnormal behaviors reported at this time consisted solely of a few episodes of self-injurious behavior.

Subject 1 received a serious wound in October of 2020 (Figure 3) and began abnormal self-directed picking of the wound.

In the management of chimpanzees with abnormal behaviors, veterinary and behavioral professionals employ preventive strategies that progress from less invasive interventions to pharmaceutical use in a gradual manner. Psychiatric medications are considered a last resort due to their sedative effects, which can impact cognitive and social functioning in these animals. This approach ensures that the most suitable and least intrusive measures are attempted before considering pharmaceutical intervention for the well-being of the chimpanzees in the social setting. In January 2021, Subject 1’s initial care plan included antibiotics as needed, pain management using an NSAID, opioids, an antihistamine, and a tricyclic antidepressant, as well as increasing the frequency of daily forage from one to three times per week providing a potential alternative behavior to his wound picking. After these veterinary and behavioral interventions, we did not see a decrease in Subject 1’s wound picking. In fact, an increase in abnormal behavior reports from 13 total reports in 2020 to 29 reports between January and May 2021 prompted further discussion of his case between the veterinarian and behavior program director.

The next stage of Subject 1’s intervention was multifaceted and included the following: antibiotics as needed; discontinuation of the tricyclic antidepressant and initiation of a selective serotonin reuptake inhibitor, Gaba analog, and Cannabidiol (CBD); relocation of the group from an open-air corral to a larger forested habitat to provide additional space to this large group; alteration of group-shifting procedures to ensure that they were not secured in a particular area for an extended time; and initiation of laser (photo biomodulation) therapy as a non-invasive therapeutic modality to decrease inflammation and pain and accelerate healing. These four interventions were instituted in May 2021. In June, we began to see a decrease in the number of abnormal behavior reports for Subject 1. In July 2021, we added an additional intervention via an increase in PRT sessions using sensory targets to distract Subject 1 from wound picking. The number of PRT sessions increased from an average of 5 sessions/month prior to the intervention to an average of 12 sessions/month afterwards; sessions during both time frames included body part presentation; however, the use of targets with different textures was implemented in July. Wound picking decreased based on TIA observations between the months of May and August 2021 (Figure 4).

Between July and September of 2021, we saw a sharp decrease in abnormal behavior (Figure 5), and although reports increased again slightly in October, November, and December they never returned to the levels observed in May.

Subject 2

After the last introduction of a male and female to Subject 2’s group in May 2023 and the removal of two females in August 2023, in October 2023, Subject 2 received a full thickness wound that was more than 1 in deep involving the right testicle (Figure 6). The subject was treated with drug therapy for pain management. Staff were unsure which group member inflicted the wound. Genital wounds tend to indicate more severe attacks [15,16], and the occurrence of this type of wound led to heightened concern regarding social group dynamics from both the veterinarian and the behavior program director.

Unilateral orchiectomy was later performed due to infection, and laser (photo biomodulation) therapy was used post surgically in conjunction with antibiotics for long-term wound management. Subject 2’s frequency and degree of wounding prompted case discussion between the veterinarian and behavior program director. The veterinarian prescribed technician-driven post-surgical laser therapy via PRT. During Subject 2’s three-month recovery period, he was pair-housed with a single female from his group to prevent re-injury from wound picking and improve access for laser therapy sessions as well as to maintain social bonds with a conspecific. During this recovery period, a geriatric male in the group passed away in December 2023; Chimp Haven often observes social hierarchy changes after the death of a group member [unpublished data]. Subject 2’s clinical case was resolved in January 2024, and he was reintroduced to his group. The behavior team was tasked with observing him via extended SMS and NNN observations. The introduction went well, and from behavioral observations, it appeared he was regaining his status as alpha. According to our NNN observations at night, which provide us with insights into social dynamics during the evening and provide information on which group members are spending time in proximity and likely have significant bonds, Subject 2 was spending time in proximity to two females with high social status (Figure 7) which led us to believe his reintegration into the group would proceed without issue.

In addition, SMS data on Subject 2 between December 2019 and March 2024 indicated little to no aggression of any type and higher levels of affiliative behavior (8.13%) than the colony average (Figure 8).

However, in March 2024, Subject 2 experienced a canine avulsion and received substantial wounding including a laceration to his back necessitating surgical intervention due to epithelial bridging and pocketing, causing concerns for infection (Figure 6). In this altercation, staff directly observed the newest male wounding him. Again, laser therapy and antibiotics were used post operatively for wound management over the course of two months. It was determined during discussions between the veterinarian and behavior program director and with staff input, that he would return to his group, and that the newest male and female introduced to the group would be introduced elsewhere. Subject 2 again became the alpha of his group.

## 4. Discussion

The fundamental nature of many nonhuman primate species, particularly chimpanzees, is inherently gregarious, thriving within large, intricate social groups in their natural habitat [17,18]. The absence of proper social housing for nonhuman primates curtails their capacity to partake in behavior typical of their species, resulting in adverse effects on their physiological and psychological well-being. Social housing provides captive nonhuman primates with a more socially-appropriate setting; although, it brings about management intricacies, including the establishment and perpetuation of harmonious social groups [19,20,21,22], ensuring access to medical care and treatment, heightening the likelihood of injuries [23,24,25], as well as the potential transmission of diseases [26,27,28,29].

This highlights the need for a strong collaboration between veterinarians and behavioral program managers to maximize animal welfare and diversify animal care and use programs to incorporate an interdisciplinary approach to address psychological well-being and physical health concerns in captive primate populations.

Chimp Haven veterinarians play a crucial role in identifying chimpanzees that are at risk for or already affected by conditions such as obesity, cardiovascular disease, and metabolic disorders. Once identified, this information is communicated to behavioral staff, who then assign a trainer to the identified animal for specific operant conditioning training, including but not limited to awake weight training on scales located in interaction medical rooms, laser therapy and KardiaMobile™ training, as well as urine and sperm collection. This approach ensures that staff continue to gather necessary diagnostics for monitoring while the chimpanzees remain within their social group. By integrating veterinary care with behavioral training, we can proactively address health issues and provide tailored individual support for the chimpanzees under our care [30,31,32].

This integrative medical care approach allows the chimpanzee to actively participate in the treatment process after the veterinary and behavioral teams have analyzed the physical, nutritional, behavioral, and environmental elements. This holistic approach not only fosters inclusion but affords maintenance of the social housing which has proven to provide positive healing outcomes and stands as the cornerstone of effective and compassionate animal care at Chimp Haven as well as at other facilities. At one facility, positive reinforcement training was used to treat a chimpanzee for airsacculitis [33], while another facility detailed a combination of medical and behavioral interventions to treat self-injurious behavior [34]. Research on the role of behavioral management combined with top notch veterinary care continues to demonstrate multiple benefits to the nonhuman primates in our care [35,36].

To support this unified medical and behavioral care approach, Chimp Haven recognizes there are many benefits of collecting and maintaining behavioral data sets for chimpanzees in managed care. This includes having clear, objective criteria when determining next steps for an individual as well as building datasets for both individuals and groups over time to allow for comparisons over the duration of long chimpanzee lifespans. In both cases covered here, the marriage of medication, monitoring, and positive reinforcement training, driven by the veterinarians and behaviorists, improved outcomes for these individuals.

Future Plans

Chimp Haven acknowledges the significance of providing medical treatment within the social group to yield the most positive therapeutic outcomes. Consequently, the organization has strategically integrated interactive medical rooms into each animal housing building. These facilities are designed to ensure convenient access for obtaining awake diagnostics with advanced medical behaviors such as venipuncture, blood pressure monitoring, awake body weights, and the use of KardiaMobile (ECG) technology.

This progressive approach enables veterinarians and behaviorists to collaboratively develop and implement individual treatment plans. The outcome of this collaboration is the acquisition of necessary long-term diagnostics without the need for anesthesia and to reduce the instances of removing individuals from their social groups. This approach is expected to become increasingly essential as the captive population ages and as the social hierarchies change within groups from both new additions and natural attrition. Organizational goals include maintaining chimpanzees in large social groups, an increase in chimpanzees trained and maintained on medical behaviors, awake weights, and emergency recalls for chimpanzee safety. All of these programs involve the support of our veterinary, behavior, and husbandry departments that make up our animal care team. By implementing these measures, Chimp Haven aims to continue to ensure the well-being of its residents by providing advanced and compassionate medical, husbandry, and behavioral care.

## 5. Conclusions

In the captive setting, chimpanzees may need to be isolated for short periods to facilitate initial healing, and groups may require re-evaluation as well as compelling careful planning and execution to ensure the continuity of care and well-being for all chimpanzees. This comprehensive model involves leveraging the expertise of veterinarians and behaviorists to create a holistic framework for promoting the well-being of chimpanzees, catering to both their psychological and physical health needs. This is based on the holistic medicine principle of healing requiring a team approach, incorporating the patient, veterinarian, and behaviorist.

Implementing this approach requires a high level of communication, expertise, and training among the animal care team, and demands time for data collection to ensure the personalization and specificity of the individual care plans. Despite the challenges, this commitment promotes highly customized and effective treatment options, ensuring the well-being and development of each individual. This model also supports the maintenance of a social housing environment, facilitating the swift integration or reintegration of chimpanzees into communal living which promotes the social interaction that is a staple for this species. Overall, these proactive management strategies are crucial to the physical and psychological health of socially housed captive nonhuman primate populations.

## Figures and Tables

**Figure 1 vetsci-11-00523-f001:**
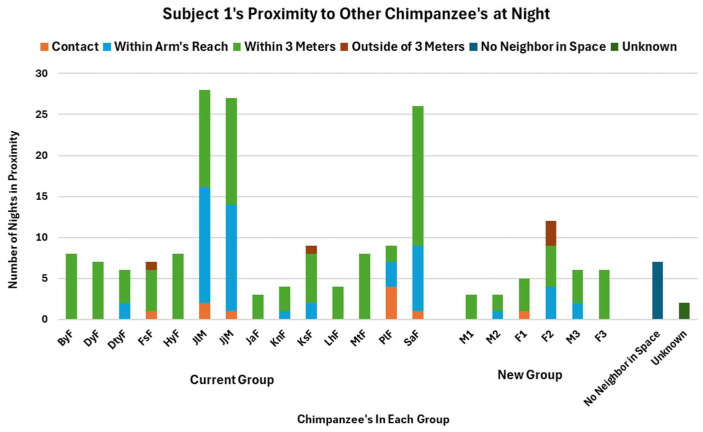
Subject 1’s proximity to his current group members and new group members during Nearest Night Neighbor (NNN) observations between August 2019 and March 2020.

**Figure 2 vetsci-11-00523-f002:**
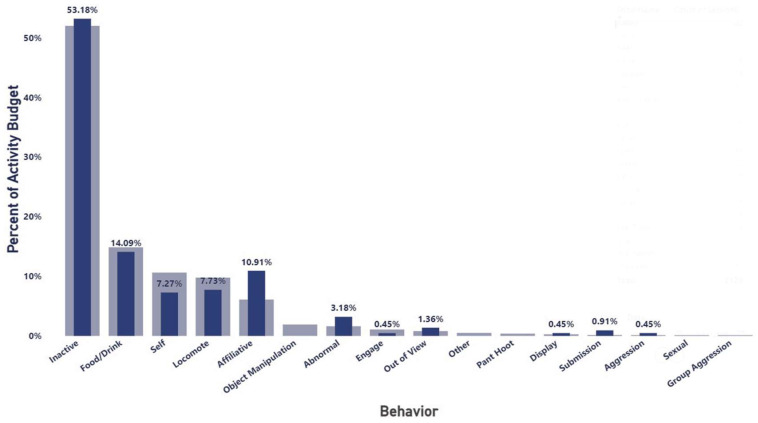
Subject 1’s behavioral time budget based on Social Monitoring System (SMS) observations between August 2019 and March 2020. Dark blue bars represent Subject 1’s behavior, while the lighter bars are representative of the colony overall.

**Figure 3 vetsci-11-00523-f003:**
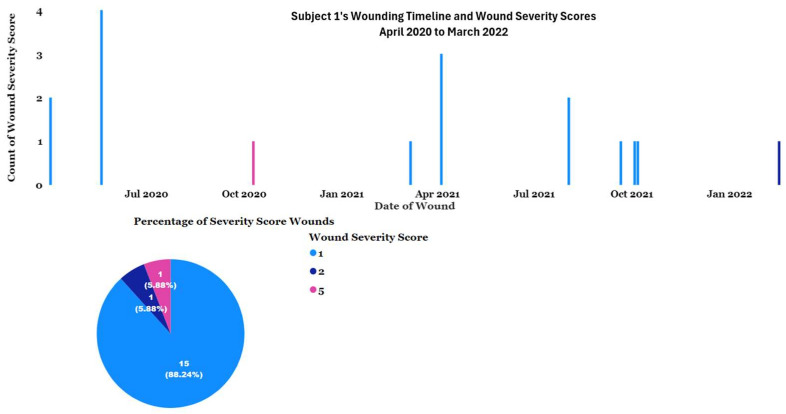
Subject 1’s wounding timeline and severity of wounds received between April 2020 and March 2022 from Wounding Reporting data.

**Figure 4 vetsci-11-00523-f004:**
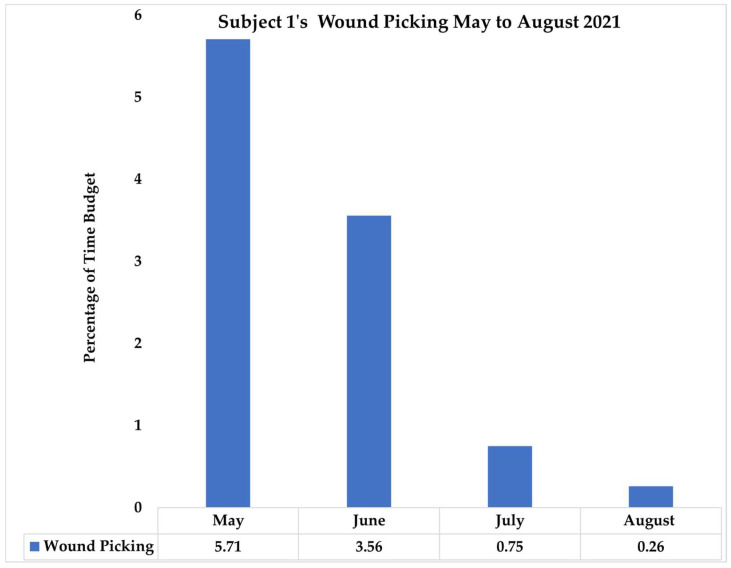
Subject 1’s percentage of behavioral time budget involved in wound picking between May and August 2021 based on Targeted Individual Assessment (TIA) observations.

**Figure 5 vetsci-11-00523-f005:**
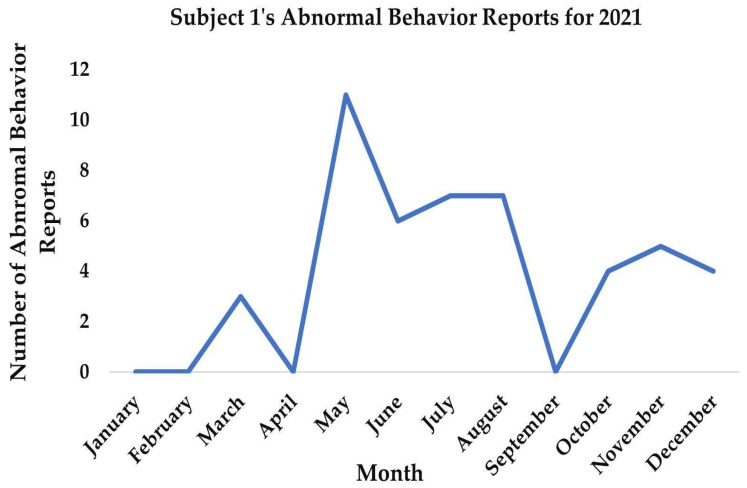
Subject 1’s abnormal behavior reports for the year 2021.

**Figure 6 vetsci-11-00523-f006:**
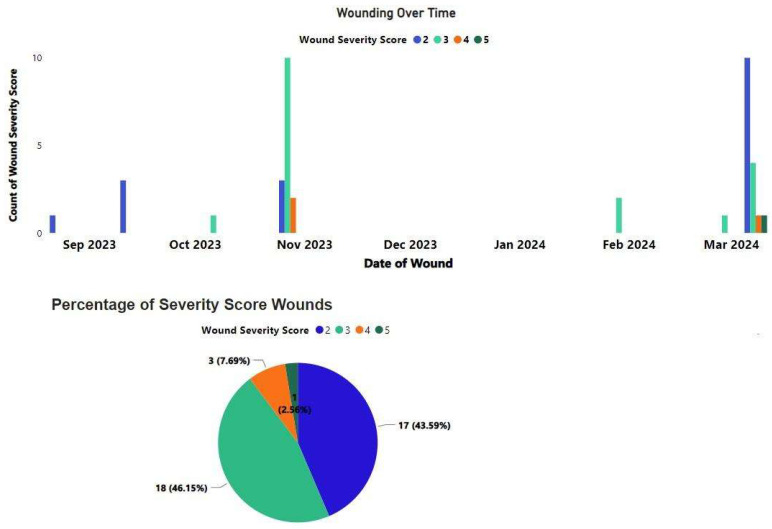
Subject 2’s wounding timeline and severity of wounds received between May 2023 and May 2024 from Wounding Reporting data.

**Figure 7 vetsci-11-00523-f007:**
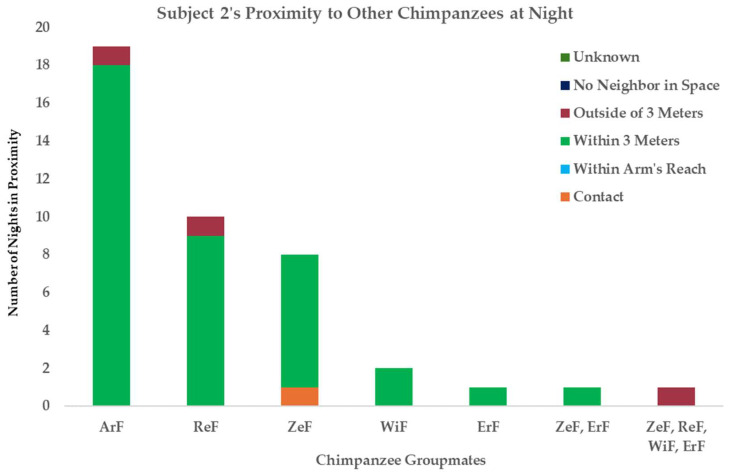
Subject 2’s proximity to his group mates from Nearest Night Neighbor (NNN) observations from January to early March 2024.

**Figure 8 vetsci-11-00523-f008:**
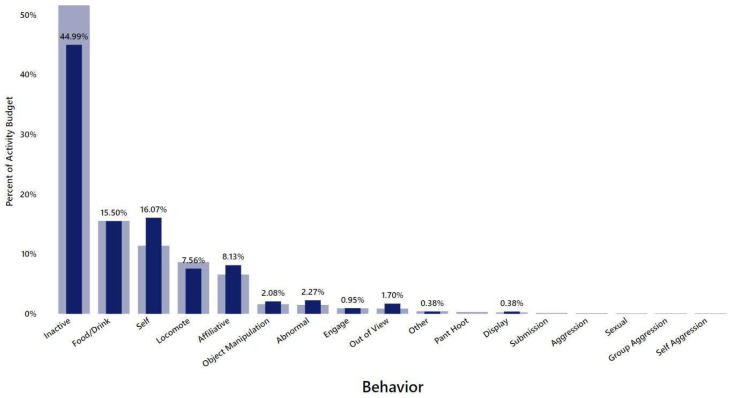
Subject 2’s behavioral time budget based on Social Monitoring System (SMS) observations between December 2019 and March 2024. Dark blue bars represent Subject 2’s behavior, while the lighter bars are representative of the colony overall.

## Data Availability

The data are contained within this study.

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
