# Peer review of "The Interface Between Veterinary and Behavioral Management of Chimpanzees (Pan troglodytes) in a United States Sanctuary Demonstrated by Two Clinical Cases"

_vetsci, 2024, doi:10.3390/vetsci11110523_

Round 1
Reviewer 1 Report
Comments and Suggestions for Authors
This manuscript explores the interface between veterinary and behavioral management of chimpanzees (Pan troglodytes) in a sanctuary in the United States.
The relevance of the topic is undeniable and deserves prominence in the scientific community, especially in high-impact journals like this one. However, the presentation of the two clinical cases that used this approach was obscured by an introduction centered on the sanctuary and a discussion about the importance of multidisciplinary themes for chimpanzee welfare.
My observations aim to encourage the authors to share the success of their practices more broadly, demonstrating the care, concern, dedication, and expertise regarding the physical and mental health of chimpanzees. Upon concluding the reading, I realized that the way the cases were presented might make it difficult for readers to identify this information from the title, keywords, and abstract.
Therefore, my suggestions would be:
-
Maintain this manuscript with the same title and introduction, but include in the results macro information about the cases addressed within the proposed interface, i.e., a general survey of the Sanctuary's management, which could be discussed in terms of success and failure percentages. Suggestions for changes in approaches and so on.
-
Publish in the format of case reports, valuing the proposed interface, but discussing other similar clinical cases in chimpanzees or primates from other institutions, such as primatology centers.
-
Adjust this article, but provide more details about the cases addressed.
-
Include in the title that it deals with two clinical cases within the proposed interface approach;
-
Bring in the introduction some information about the clinical issues presented in the two cases to be discussed;
-
Discussion with articles related to the chimpanzees (individuals) in question;
-
Conclusion more focused on the success of the interface – in a more succinct and direct way with the objective of the work.
I reiterate that my rejection of the article is not due to its quality or the information presented, but rather to how the data was organized. I believe that a new presentation and expansion of such relevant data will be better valued and accessed by readers and professionals in the field, serving as a reference for excellent behavioral management carried out at Chimp Haven.
Reviewer 2 Report
Comments and Suggestions for Authors
no significant edits or comments
Author Response
Comment 1: Comments and Suggestions for Authors no significant edits or comments
Response: We would like to thank Reviewer 2 for their review of our article and share our appreciation for their time spent reading and reviewing it for us.
Reviewer 3 Report
Comments and Suggestions for Authors
Comments and Suggestions for Authors
The authors used multiple methods of data collection including but not limited to focal continuous observations, instantaneous and all occurrence observations, staff surveys, and anecdotal input from animal care staff. Why were paraclinical test imaging (magnetic resonance imaging) not carried out to analyze the deterioration of the brain and find out if they are linked to the behavior of chimpanzees? Laboratory tests to determine if there was deterioration of the immune system, after being subjected to different biological agents (BH), hormonal profile, etc. Because the chimpanzees are frequently caged alone and deprived of the freedom, autonomy, and meaningful social interaction essential to their well-being. As a result of enduring the terror and pain of having their bodies routinely violated for experiments and the loneliness of their tiny steel and concrete prison cells, many chimpanzees bear lifelong emotional scars. Numerous studies have shown that even long after they've been retired from experimentation, many chimpanzees exhibit abnormal behavior indicative of depression and post-traumatic stress. They suffer from symptoms such as social withdrawal, anxiety, and loss of appetite. They pull out their own hair, bite themselves, and pace incessantly.
It is suggested to resubmit your manuscript to the same journal as a case report.
Author Response
Summary:
Thank you very much for taking the time to review this manuscript. Please find the detailed responses below and the corresponding revisions/corrections highlighted/in track changes in the re-submitted files |
Comments 1: The authors used multiple methods of data collection including but not limited to focal continuous observations, instantaneous and all occurrence observations, staff surveys, and anecdotal input from animal care staff. Why were paraclinical test imaging (magnetic resonance imaging) not carried out to analyze the deterioration of the brain and find out if they are linked to the behavior of chimpanzees? Laboratory tests to determine if there was deterioration of the immune system, after being subjected to different biological agents (BH), hormonal profile, etc.
Response 1: Thank you for this inquiry. We have included additional information to the "Housing and Care" section to provide this information.
Comments 2: Because the chimpanzees are frequently caged alone and deprived of the freedom, autonomy, and meaningful social interaction essential to their well-being. As a result of enduring the terror and pain of having their bodies routinely violated for experiments and the loneliness of their tiny steel and concrete prison cells, many chimpanzees bear lifelong emotional scars. Numerous studies have shown that even long after they've been retired from experimentation, many chimpanzees exhibit abnormal behavior indicative of depression and post-traumatic stress. They suffer from symptoms such as social withdrawal, anxiety, and loss of appetite. They pull out their own hair, bite themselves, and pace incessantly.
Response 2: The authors acknowledge that past negative experiences often have long term consequences on both the physical and mental states of the chimpanzees in our care. In many cases, with limited historical information, we may not be able to determine the direct causes of either behavioral or physical issues.
Comment 3: It is suggested to resubmit your manuscript to the same journal as a case report.
Response 3: The authors thank Reviewer 3 for their time spent reviewing our manuscript as well as their questions and input. After careful consideration, we have decided to indicate in the title, abstract, and keywords that this manuscript deals with two specific clinical cases rather than resubmitting as a case report. We were invited to submit to a special issue specifically about the interface between veterinary and behavioral care which is our focus.
Reviewer 4 Report
Comments and Suggestions for Authors
This is a valuable manuscript that demonstrates the important interconnection between behavioral management, behavioral monitoring, and veterinary medicine in fostering improved welfare, social outcomes, and overall health in socially-housed chimpanzees. It shows how various behavioral management- and veterinary health-related monitoring and metrics can provide a variety of information about animal status to guide data-driven welfare decisions, and discusses two example cases highlighting05! the value of these complementary areas. Overall, the manuscript emphasizes the important information that behavioral management data can provide in guiding social- and care-related decisions, encouraging its application in the care of socially-housed captive primates.
I have including the following comments and suggestions to help strengthen the manuscript:
Lines 100-106: For wound reporting, could you please include some details about criteria used to score wound severity? I think this would be helpful to the readers especially since wound scoring/reporting comes into play later in the results sections for both subjects.
Lines 107-109: When monitoring abnormal behavior, are there specific behaviors/stereotypies that are measured, or is the reported abnormal behavior more about any behavior that is atypical in the context of a given individual’s usual behavioral repertoire?
Lines 110-113: I think it would be valuable for the readers to have a little more insight/detail about the implementation of the social monitoring system, which seems like such an important tool for recognizing welfare-related changes in the animals. How often is the social monitoring system used to gather baseline activity budget data - Is it performed for individuals on a weekly, monthly, or yearly basis, or as needed? When observations are performed, are they performed at all times of day, and for what typical lengths of time?
Lines 128 – 133: Similarly, the positive reinforcement training is such a valuable tool for helping to facilitate care as well as providing a form of cognitive enrichment on its own, and it may be helpful to elaborate slightly on this program here - About how frequently are individual animals trained (multiple times a week, weekly, monthly)?
Lines 162-163: There is a minor mismatch here in the numbers described. Text reads “maintained in a large 21-member multi-sex social group (n=15…)”. Please correct for consistency.
Lines 225-226: Similar to the comment regarding lines 107-109, are there any examples of what is typically classified as “abnormal behavior” in Subject 1’s behavioral time budget.
Lines 262-263: For Subject 1’s described increase in PRT sessions, how frequently on average were PRT training sessions implemented both before and after the increased intervention? Were multiple PRT training procedures increased, or just training of one specific skill?
Lines 333 – 345: These were great paragraphs highlighting to the audience why social housing is so essential despite its inherent risks in the captive setting, and how all of these behavioral programs can help mitigate and deal with these risks to the benefit of overall welfare. Could consider adding a similar sentence to go along with lines 45-46 (talking about how chimpanzees live in large social groups) in the introduction to help emphasize from the start why this is important for chimpanzee welfare.
Round 2
Reviewer 1 Report
Comments and Suggestions for Authors
The requested changes have been addressed by the authors, and I believe the manuscript is suitable for publication.